# An Algorithm with Iteration Uncertainty Eliminate Based on Geomagnetic Fingerprint under Mobile Edge Computing for Indoor Localization

**DOI:** 10.3390/s22239032

**Published:** 2022-11-22

**Authors:** Jie Li, Liming Sun, Dongpeng Liu, Ruiyun Yu, Xingwei Wang

**Affiliations:** 1School of Computer Science and Engineering, Northeastern University, Shenyang 110169, China; 2University of Texas at Dallas, Richardson, TX 75080, USA; 3Software College, Northeastern University, Shenyang 110169, China

**Keywords:** indoor localization, multisensor fusion, magnetic fields, sensors, Kalman filter, PDR, edge computing

## Abstract

Indoor localization problems are difficult due to that the information, such as WLAN and GPS, cannot achieve enough precision for indoor issues. This paper presents a novel indoor localization algorithm, GeoLoc, with uncertainty eliminate based on fusion of acceleration, angular rate, and magnetic field sensor data. The algorithm can be deployed in edge devices to overcome the problems of insufficient computing resources and long delay caused by high complexity of location calculation. Firstly, the magnetic map is built and magnetic values are matched. Secondly, orientation updating and position selection are iteratively executed using the fusion data, which gradually reduce uncertainty of orientation. Then, we filter the trajectory from a path set. By gradually reducing uncertainty, GeoLoc can bring a high positioning precision and a smooth trajectory. In addition, this method has an advantage in that it does not rely on any infrastructure such as base stations and beacons. It solves the common problems regarding the non-uniqueness of the geomagnetic fingerprint and the deviation of the sensor measurement. The experimental results show that our algorithm achieves an accuracy of less than 2.5 m in indoor environment, and the positioning results are relatively stable. It meets the basic requirements of indoor location-based services (LBSs).

## 1. Introduction

The demand for indoor positioning technology is becoming more and more important in recent years. With the popularity of wireless communication technology and the rapid development of intelligent terminals, wireless positioning technology has become a popular research focus [1]. Recent advancements in the Internet of Things (IoT) have led to the emergence of new applications, one of which is positioning, commonly known as localization. Localization, in its simplest terms, is the process of making something local to an area, which can be achieved through the use of indoor positioning systems (IPSs) and location-based services (LBSs) [2]. Many applications which build on LBSs require a precise indoor location to improve their service quality, for example, the smart shopping mall, smart museum’s navigation for tourists, indoor navigation from fire zone to safe zone, etc. [3]. Automatic mobile robots are also widespread; the industrial operation vehicle has been widely applied [4], which can be a good example. In addition, the household robot cleaners are providing reliable services for families, which also facilitate the gradually increasing LBS market. Moreover, indoor localization technology also plays an important role in elderly care, disaster management, and assistance [5]. COVID-19 has been spreading globally since December 2019. Indoor localization technology can effectively monitor the trajectory of infected persons and realize the contact and track tracing of infected persons, which is of great significance for the implementation of precise prevention and control policies [6]. Indoor location-based services have drawn much attention in the past decades because position information is essential for providing intelligent services in various fields in the context of Internet of Things [7]. Many IoT applications will require seamless and ubiquitous indoor/outdoor localization and/or navigation of both static and mobile devices. Traditional short-range communication technologies can quite accurately estimate the relative indoor location of an IoT device with respect to some reference points, but the global location (i.e., longitude–latitude geographic coordinates) of these devices remains unknown, unless the global location of the reference points is also known. Emerging long-range IoT technologies can provide an estimate of the global location of a device (since the exact locations of their access points are normally known); however, their accuracy is deemed very low, especially for indoor environments [8]. In order to provide better indoor services, the navigation system needs to know the precise location of humans or devices [4]. Although the Global Positioning System (GPS) is a good choice for outdoor localization, it cannot provide the required LBS for indoor environments. This is because in most indoor environments, the concrete structure blocks the GPS signal, so GPS is not able to provide indoor positioning with satisfactory accuracy [9]. Indoor localization technologies still need to be more efficient and reliable to guarantee the positioning under complex building structures [10]. A number of indoor localization systems have been proposed in recent years [11,12,13], which use different techniques [14].

Recently developed technologies for indoor navigation can be divided into two categories [4]. One category uses absolute positioning, and the other uses relative positioning. In general, absolute position is estimated using external devices and signals, such as ultrasonic [15], WiFi [1], Bluetooth [16], and radio frequency identification (RFID) [17]. Relative positioning, such as an encoder system and an inertial navigation system (INS) [18], uses sensors to measure motion and relative position. These sensors can also be installed in mobile devices and robots. Based on these sensors, a variety of indoor positioning technologies have emerged.

In absolute positioning, wireless fingerprint-aided positioning methods are popular. They focus on improving the positioning accuracy as well as real-time performance [19]. To be specific, fingerprint-aided methods record the signal fingerprints of each position and store it in the fingerprint database for future matching. From previous works we can conclude that there are two main ways to improve the fingerprint positioning accuracy: by increasing the fingerprint density [20], and by improving the positioning algorithm or model [21]. However, increasing the fingerprint density will reduce the real-time performance of systems, because in this case, both the direct interpolation location [22] and the probability distribution location [23] need more fingerprint matching time. In other words, when running on a limited hardware environment such as a mobile device, the time consumed by fingerprinting methods cannot meet the requirements of real-time localization. In this case, before fingerprint matching, we can employ a K-means clustering algorithm [24] to improve the positioning process. The clustering method is used to ensure the accuracy of the system, greatly reducing the positioning time and effectively improving the real-time performance.

In addition, some signals such as WiFi and Bluetooth may be affected by attenuation, multipath fading, and human obstruction [1], which may result in a decrease in positioning accuracy. Another drawback is that some absolute positioning systems using these signals require some hardware infrastructure, such as routers or beacons, to be installed in the building [24]. Taking all these problems above into consideration, we need to find a stable and inexpensive signal for mobile devices. The geomagnetic field is refracted by structures such as pillars, steel structures, and fixed large objects. The fingerprint map navigation with geomagnetism is able to estimate absolute position without infrastructure. Therefore, compared with the traditional WiFi and RFID signals, magnetic signals are suitable for indoor positioning.

On the other hand, a relative positioning system does not require a complex hardware infrastructure, as the required sensors have been widely installed in the mobile device, for, the gyroscope and accelerometer. Positioning methods based on these sensors are popular among recent studies, as these sensors are available everywhere, which means they are fairly user-friendly [1]. A widely used relative positioning method with these sensors is pedestrian dead reckoning (PDR), to measure the user motion pattern and finally find the walking trajectories or locations. Current position would be determined according to the previous position of pedestrians [25]. Some location-related factors of PDR, including the initial point, the detected step length, and the walking direction, will affect the accuracy of the positioning result [26]. In addition, different implementations of PDR have a different form of these factors.

However, PDR can empirically provide high positioning accuracy in short distance. This is because sensor drift is a common problem in the PDR system [27]. When estimating distance and direction, inertial sensors always have some small measurement errors and signal noise, such as slight body movement, which will further exacerbate this problem [4]. It drifts along the walk distance because the method would use the relevant location information. In other words, the longer the user walks, the worse the estimation will be. PDR and other relative positioning methods commonly suffer from sensor drifts; thus, they may have divergence problems due to cumulative error. Thus, some PDR systems use a map-matching mechanism to calibrate these errors [28]. To solve this problem, some recent studies combined fingerprint-based absolute positioning and PDR method, to make them complement each other, reducing cumulative error and achieving better positioning accuracy. For example, attitude and heading reference systems (AHRSs) fuse observation from inertial sensors and magnetic sensors for accurate orientation determination [29]. A simple and effective fusion algorithm is needed in order to achieve real-time implementation on a limited-resource mobile platform. Filter algorithms, such as Kalman filter, are commonly used in tracking and localization topics, as they can easily fuse the observation and prediction [9]. Using Kalman filter to combine PDR and fingerprint map positioning involves two steps. The first step leads to position prediction, and this primary prediction could be updated at the second step, with the help of fingerprint map positioning and PDR. GeoLoc applies the lightweight Kalman filtering algorithm for fusion estimation of fingerprint map matching and PDR.

Although a lot of work has been carried out in this area, some common critical issues still need to be explored and resolved to enhance the accuracy and effectiveness of our method. Problems are mainly regarding cold start [30] and error accumulation [15], which are also the common causes of positioning errors. The cold start problem means that the positioning results are inaccurate at the beginning of the positioning, and the accuracy will be improved with the increase of walking distance. The reason for cold start problems is that the algorithm cannot be corrected without the location information of the previous steps. By using fusion with fingerprint positioning, we find that cold start problems can be avoided to some degree, especially at the beginning stage of the positioning. Error accumulation often occurs in the later stage of trajectory tracing, and both measurement data and inappropriate parameters of method would cause positioning failure. In terms of data sources, the sensor measurement noise will lead to large cumulative error; thus, the location error would be increased at the later stage of positioning.

The indoor localization technology mentioned above provides high-precision positioning, but the complex localization algorithm requires more processing time and increases the power consumption of the equipment. On the other hand, it also faces great challenges in some time-delay sensitive scenarios. Traditional cloud computing provides powerful service capabilities such as storage, memory, and computing power for IoT devices. However, due to the long distance and high delay, the system reliability is weak. Due to the centralization of the cloud server, the system has single point of failure and network congestion. Cisco has come up with the concept of MEC, which can scale the computing resources of the cloud located at the edge of the network to solve the problems of insufficient computing resources and latency in indoor positioning [31].

In order to solve the above problems and improve positioning performance, GeoLoc proposed an iterative uncertainty elimination algorithm that combined magnetic field map navigation and Kalman filter PDR. Its main contributions are as follows:The uncertainty of candidate position estimation is proposed to solve the cumulative error problem. The experimental results show that GeoLoc can successfully reduce the cumulative error.Inspired by particle filter, this paper introduces the concept of path candidate set to enhance the robustness of localization. To guarantee fault tolerance, GeoLoc employs adaptive location candidate set for magnetic fingerprint matching.Since GeoLoc only applies the sensor of the mobile phone, no other hardware needs to be deployed, and it has the advantages of low time complexity and self-regulation.To solve the cold startup problem, users only need to walk a few steps to obtain accurate positioning results.

The organization of the rest of the paper is as follows: Section 2 lists previous localization models. In Section 3, we explain our approach and model architecture. Section 4 presents the implementation and experiment results. Section 5 is dedicated to the conclusion.

## 2. Related Work

Many methods have been proposed for indoor localization over the past decade. Our work is based on previous studies of fingerprint map matching, step length estimation, and filtering positioning.

Commonly applied indoor positioning technologies make use of a wide range of sensor data, from methods using RFID [17] or WLAN [32] to SLAM-based [33] systems. P. Bahl et al. [34] proposed a radio frequency (RF)-based system using consuming infrastructures. In addition, infrared [35], Bluetooth [36], and ultrasonic technology [16] have been applied in indoor positioning systems. Ref. [37] used GSM signal positioning, but the precision can only reach 10 m, which cannot meet the requirements of indoor positioning environments. Although some of these methods with corresponding signals have achieved relatively high accuracy, they all rely heavily on infrastructures. However, because of energy consumption of infrastructure, they are difficult to use in practical problems, and they may be ineffective when signals are weak. Therefore, indoor positioning methods that are not dependent on infrastructure, for example, mobile-device-sensor-based systems, are widely accepted.

In recent years, machine-vision-based localization methods have also been extensively studied. This method is made available by using camera data of a mobile device. The 3D model of machine vision can be constructed from the visual aspects of the interior; thus, theoretically, high positioning accuracy would be achieved [38]. However, the application of professional camera equipment is expensive. To process pictures, a large amount of computing resource is needed as well, so it is not suitable for common indoor scenarios with a mobile phone. The image location method is simple and easy to implement, but it is difficult to calculate the distance between objects because of the lack of depth information. Only an auto focus system can solve this problem [39].

Geomagnetic field is a popular data type selection for fingerprint map. Because of the influence of the steel frame structure on the indoor magnetic field in the modern building, the inhomogeneous indoor magnetic field is formed, resulting in the difference of the magnetic field in different positions. Therefore, it is possible to make use of the uniqueness and stability of the magnetic field in the indoor environment. Some studies have begun to explore geomagnetism for positioning. Magnetic indoor local positioning systems (MILPSs) [40] have been proposed. MILPSs provide accurate and reliable artificially generated magnetic field maps, which cover the most basic infrastructure, and use the fewest facilities to complete the indoor positioning system of the whole building. These solutions show great prospects. With the development of artificial intelligence, ref. [41] proposed a magnetic-based indoor localization system using deep learning technologies, which uses a robot carrying a smartphone to collect magnetic field data in an indoor environment. Vision-based technology is used to track the position of the robot and use a neural network to automatically understand the relationship between magnetic field readings and position. The technology can potentially be integrated with other global indoor positioning systems to improve accuracy. The geomagnetic fingerprint-based method has high positioning accuracy, but it needs to establish the fingerprint database or map in advance. Through comparison, GeoLoc uses geomagnetic data after the deflection angle transformation to construct the geomagnetic fingerprint map.

In some fusion systems, map-matching mechanisms play a role [25] to calibrate the sensor errors caused by PDR. There are two ways to calibrate errors. The first attempt is to match user trajectories to the nearest intersection and road on the map, while the other way uses map information to filter out locations that users are unlikely to walk, such as walls and obstacles. Both techniques require a detailed scaled map of the building. Carlos E et al. [42] described the necessity of developing a geomagnetic fingerprint map, and presented an improved positioning method on a smartphone platform. Because the computing resources of mobile devices are very limited comparedto computers, the main purpose of this work was to reduce the amount of data stored and processed. To achieve this, a genetic algorithm (GA)-based [10] method was proposed by them, to select the feature and find the location of the user. This algorithm achieved good results in the experiment but it is time-consuming.

ChengKai Huang [43] proposed a magnetic map-matching algorithm and an improved pedestrian dead reckoning algorithm (IPDR). This improved pedestrian dead reckoning provides efficient indoor navigation. In the work of Sheng Guo [44], using geomagnetic sensor signals for indoor positioning, a novel multilevel link node model containing geometric and topological information was proposed to construct a magnetic field strength (MFS) sequence fingerprint database. By using this database, a combination of dynamic time warping (DTW), pedestrian dead reckoning (PDR), and k-nearest neighbor (kNN) hybrid methods were usedto achieve effective MFS sequence matching and accurate indoor positioning. The above two methods are effective ways to obtain and use the built-in sensors of the smartphone for different indoor navigation. Therefore, they require less infrastructure cost to implement. In their work, the step and direction are estimated using magnetometer and gyroscope sensors to find the user’s movement. However, sometimes sensors are affected by metal objects in the indoor environment; in other words, the gyroscope sensor will also be disturbed by the sensor drift problem [10]. Sometimes the IPDR method cannot provide accurate location for users; therefore, some methods use filter methods to smooth the observation. Ref. [45] presented a novel map-matching method using rasterized maps. By modifying the recursive cycle of the Bayesian filter, the computationally efficient map filter can be designed as an alternative method for particle filter map matching. The introduced map is also used to generate scores for a multiple hypothesis testing (MHT) approach that can find the correct initial position and heading of the pedestrian trajectories. While the experimental results serve as a proof of concept for the designed map filter and the self-initialization method, they do not prove the correctness of pedestrian long-distance walking.

In some scenarios, particle filter [46] has a fair result, but the time complexity of it is another severe problem. As a nonlinear and non-Gaussian estimation, particle filter sometimes has a better performance than Kalman filter, but in indoor localization environments, Kalman filter methods play a role since the indoor motion models of the user are relatively simple and can be transformed to linear and Gaussian formats [47]. A method of self-positioning magnetic field mapping based on particle filter is studied to solve the cumulative error problem by introducing uncertainty estimation [48]. The specific mode of particle propagation in these systems is fixed, such as the speed limitation. The particle motion would not depend on the help of other positioning systems. However, if the movement of a moving object is not in a fixed position, the estimation of particle propagation is inaccurate and inefficient. The time complexity of particle filter positioning also cannot meet the real-time requirement. Ref. [49] utilized KF algorithm for sensor fusion; the study fused the accelerometer and gyroscope data for pitch and roll estimation, and fused magnetometer and gyroscope data for heading estimation. The study used the complementary features of two kinds of sensors and fundamentally addressed the accumulated errors that exist in PDR; the experimental results showed that it achieves better performance. Ref. [50] proposed a fusion algorithm combining EKF (extended Kalman filter) and PF (particle filter) to fuse the information obtained from the PDR module and the magnetic fingerprint module. The proposed algorithm can improve the inherent blindness and solve particle degradation issues in the traditional PF scheme. Furthermore, reducing the number of particles had little impact on the accuracy of the fusion algorithm. Thus, we examined some popular methods using Kalman filter with particle filter in the Experiments section.

GeoLoc implements indoor positioning around smartphone sensors without any requirements of the external equipment. In terms of the fault tolerance of positioning methods, many methods cannot tolerate irregular motion, such as the change of the moving direction, because they will introduce noise and random errors. GeoLoc, with the aid of the transformed magnetic fingerprint map, can avoid some errors caused by the random changes of direction of the mobile phone. As a result, mobile users can use their own mobile phones while using location services and walking as usual. Because GeoLoc partially tackles the partial cumulative error and cold start problem by combining PDR and fingerprint matching, it has an accurate estimate for both short- and long-distance walking. In addition, GeoLoc also uses a new dynamic motion model to improve the PDR; thus, the tracking results are better than the vanilla-PDR-based methods and the Kalman-filter-based methods. Then, we propose an iterative uncertainty elimination algorithm for GeoLoc. First, we use the geomagnetic data to match a certain range of positions. The weight of every candidate position point in each region is evaluated. Then, based on the data of the accelerometer and the gyroscope, the user’s step model, including step size and direction estimation, are also obtained. By modeling the noise (or uncertainty) of the sensor measurement, measurement error is reduced and confidence degree is increased. In the candidate set, some low probability positioning results under the threshold will be filtered out, and through the feedback of position estimation results at previous steps, the direction and step size estimation will be updated and become more precise by iterative adjustments, so the uncertainty and error of these estimates will gradually decrease. Finally, GeoLoc uses the Kalman filter to smooth the track result and the result path (or path set) of the geomagnetic matching to obtain the final location result. As shown later, in the Experiments section, GeoLoc has good performance and can meet the needs of indoor positioning.

## 3. Algorithm Design

We divided the localization process into procedures, including matching, filtering, and orientation estimation. Formula (Equation 1) shows vector *X*, the state of GeoLoc, including position vector, position *P*, and angle vector orientation *O*.
(1)$$X={\left[P,O\right]}^{T}$$

Our final target is approximating the real trajectory of the user. First, we collect geomagnetic data, then build a map for matching in offline phase. Because of the low discernibility of magnetic matching, the user should take some steps to obtain the final position, and the fewer, the better.

First, we construct a transformed magnetic fingerprint map as follows, and a sequence of data, including magnetic and gyroscope recording. For each step in trajectory data, GeoLoc will match the magnetic value of the step with the help of the transformed magnetic map, then execute optimizing the estimation, and make the decision of orientation. GeoLoc assigns elementary estimation of the next step, using a fusion of matching results and transforming of orientation with previous position. Then, after all steps have their intermediary estimation, we filter the trajectory on every step repeatedly. Finally, we find the closest trajectory to the real user trace. The model structure is shown in Figure 1.

We collected magnetic signals on the fifth and sixth floors of the information building at Northeastern University in China. We painted a grid of 40 × 40 cm on the floor. Inside each cell in the grid, we collected fingerprints at a different sampling rate, rotating the phone at a 45-degree angle to capture samples in all directions. Magnetic intensity values of uncollected locations were estimated by bilinear interpolation. We collected the magnetograph data 10 times over two months, and took the average value of the magnet at each location as the true magnet value at the corresponding location. The test phones were Samsung Note 4 and Meizu Note 6. By default, the Meizu phone will be our follow-up test tool. In Section 3, we verify two fingerprint construction methods, respectively. After verification, the Euler angle of the triaxial accelerometer is projected to the direction of gravity.

### 3.1. Transformation of Magnetic Fingerprinting with Z-Axis Projection

A built-in three-axis magnetic field sensor is common in modern mobile phones, and provides us with a convenient way to record magnetic value. This kind of sensor detects magnetic field in *x*, *y*, and *z* directions, where X points to the geographic north pole, Z represents the opposite direction to gravity, and Y is orthogonal to both X and Z. The sum of squares as total magnitude is calculated, shown in Formula (Equation 2).

Due to the difficulty of measurement of the continuous magnetic variation, our map building method firstly splits the fingerprint database into a grid according to its location. Then, we focus on finding the factors that influence the magnetic field readings in every location grid. As is generally known, magnetic field readings are quite stable over time, but would also change significantly within an limited range.Figure 2 elaborates an intuitional example where some positions may have a similar magnet value, which means that duplication is a common phenomenon in the magnet value matching process. It is unavoidable that the misjudgment of magnetic observation exists.

Because all information from three axes are used, it prevents us from imposing constraints on the orientation or placement of the phone, even if the phone is placed in a trouser pocket, a bag, or just handheld. Ref. [51] verified that this method is efficient under different occasions.
(2)$$\left|m\right|=\sqrt{x^{2}+y^{2}+z^{2}}$$

Formula (Equation 2) would guarantee the orientation to be error-free during fingerprint map construction. However, this method would give more duplicate magnetic location information. The elements in each fingerprint point will drop from three to one, reducing the uniqueness of each fingerprint, and the result is that the magnetic values will be all positive. Thus, this kind of projection will bring us an unbalanced geomagnetic map. In large indoor environments, the fingerprint positioning may need more time to converge to the right location [52].

Another method to record magnetic values is implemented with the aid of an accelerometer [53]. Because the irregular orientation of the phone is recorded by the accelerometer in IMU, we can use the offset of the three-axis data generated Euler angle to project irregular orientation to the real value. Due to the bias that exists in the readings of the sensors, estimation of the angle of rotation with respect to magnetic north pole always contains errors. Thus, a complete 3D magnetic reading along the world’s real reference frame cannot be accurately calculated. The idea is that we can employ other sensors’ information to adjust the *Z*-axis direction. The rotation angle with respect to the world’s *Z*-axis can be estimated by sensing the direction of gravity [53]. The magnetic recording along the world’s (reference frame’s) *Z*-axis can be obtained by projecting the 3D magnetic readings onto the world’s *Z*-axis provided by the gravity vector. Expression (Equation 3) shows this idea [54], employing the offset of three-axis data generated Euler angle to project irregular orientation to the real value.
(3)$$m_{z}=\vec{m}⊙\left(-\vec{g}/\left|\vec{g}\right|\right)$$
where $\vec{g}$ is the gravity vector and $\vec{m}$ is the magnetic vector measured by phone. The transformed *Z*-axis data employ Euler angle as projection of magnetic *Z*-axis data.

Note that if the biases are removed, both measurement methods work reliably. Otherwise, in the presence of sensing bias, the projection onto the *Z*-axis will be interfered with by a constant offset. However, the magnitude value always depends on a signal value among x, y, and z, due to the fact that the magnitude function (sum of squares) is not a linear function with respect to the readings. That is the reason why this feature is more reliable than calculating the magnitude of the 3D readings [55].

When this transformation is employed in map construction, we cannot directly use step magnetic observation in the matching phase. That means that the magnetic map and the step trajectory data should apply the same transformation in the same test case. Comparison experiments using two kinds of magnetic map show that the transformed fingerprinting method has a better performance, especially in short-distance localization.

### 3.2. Estimation with Eliminating Uncertainty

Quantifying the uncertainty, in this case the measurement of error distribution, is an essential way to reduce localization error. The fingerprint map-matching model makes precise revisions of uncertainty based on new data we gathered, then the following matching model is proposed.

Here, we assume that the possibility of fingerprint matching results should follow a certain distribution. In the first step, we just assign the matching result a random position that follows Gaussian distribution in the map, shown withprobability density function (PDF) (Equation 4).
(4)$$\begin{array}{cc} \; & f(P)\\ \; & =\frac{1}{\sqrt{{\left(2\pi \right)}^{k}\left|\Sigma \right|}}exp\left[-\frac{1}{2}{\left(P-\mu \right)}^{T}\Sigma^{-1}\left(P-\mu \right)\right] \end{array}$$
where μ denotes the exact matching result; in other words, the most possible position by observation. Σ represents covariance matrix, which would decide radius and inclination in uncertainty boundary. The uncertainty boundary of one step forms an ellipse area, with the major (long) axis pointing to the next possible orientation. That means that matching results take the past and future orientation into consideration. In the following steps, the mean and variance, in other words, the orientation and radius of our possibility ellipse, will be adjusted iteratively by fusion of estimation, including magnetic observation and motion model.

Figure 3 figuratively describes the boundary of a circle-shaped possibility field, formed by candidate points in the magnet map. The resulting area would be presented as several 2D Gaussian distributions. By considering a combination of all of the possibilities, we finally obtain the conjunction distribution as posterior.

Note that in the beginning, model estimation is not available, so at first, we define a specific number of start points nearest to the initial magnetic observation, which constrain the maximum number of possible paths in our result.

With the user walking, a common problem in long-distance matching is that in one step there may be no points that meet the magnetic requirement and orientation requirement. That means that this position or path is no longer reasonable for the next step’s estimation and prediction. We can tolerate the error of that path by adding a threshold, or just discarding the single path, to keep the correctness of the resulting set paths.

The next phase is choosing a method to correct the estimation we obtained in the previous phase. Filter methods, such as Kalman filter, are a common choice for smooth tracking and correcting estimation. Results and initial observation by magnetics will iteratively influence motion model. GeoLoc adopts the Kalman filter method combined with PDR and geomagnetic matching as prior knowledge to continuously update the estimation of the current state.

Then, we need to define coefficients of the Kalman filter first before we use it. The transform matrix and predict matrix here are given by experience. According to our motion assumptions, the speed of the user is several constant values, so the next prediction of our linear system is given by prediction in k−1 and velocity.

Another factor in the filter algorithm is external influence, for example, acceleration motion caused by the outside world. As a parameter in the Kalman filter, we just list it and assign it a zero vector with our assumption.

The goal of using the Kalman filter is that we want to obtain as much information from our measurements as we can. Covariance matrix *P* reveals the implicit relationship among variables. In our case, taking uncertainty into consideration and uncertainty in the previous step, updating of *P* willbe described in the following formula.

In the beginning, we assign a random constant value to *P*, because we should let *P* finally converge to an domain-related constant, in this case, user position. The next task is refining the estimation with new measurements. The transformation from sensor reading to track state is modeled by an observation matrix *H*, as presented in Equation (Equation 5):(5)$${\vec{\mu }}_{\mathrm{expected}}=H_{k}{\hat{x}}_{k}\Sigma_{\mathrm{expected}}=H_{k}P_{k}H_{k}^{T}$$

Equation (Equation 6) shows that the at *k*th step, the current state vector $\hat{x}$ is predicted from the state of the last moment with transformation *A*, coupled with the input from the outside world, which is Bu. Empirically, *B* is a zero vector.
(6)$${\hat{x}}_{k}^{-}=A{\hat{x}}_{k-1}+Bu_{k-1}$$

The prediction process increases the new uncertainty *Q*, coupled with the previous uncertainty, presented in Equation (Equation 7):(7)$$P_{k}^{-}=AP_{k-1}A^{T}+Q$$

Both models of prediction and observation have uncertainty following Gaussian distribution. By combiningGaussians we obtain the Kalman gain matrix *K* in (Equation 8):(8)$$K_{k}=P_{k}^{-}H^{T}{\left(HP_{k}^{-}H^{T}+R\right)}^{-1}$$
where *R* is also the noise of GeoLoc. We balance the prediction and observation using this gain; on top of that, we update *P*, which represents the uncertainty in this step, shown in (Equation 9) and (Equation 10):(9)$${\hat{x}}_{k}={\hat{x}}_{k}^{-}+K_{k}\left(z_{k}-H{\hat{x}}_{k}^{-}\right)$$
(10)$$P_{k}=P_{k}^{-}-K_{k}HP_{k}^{-}X_{k}$$

${\hat{x}}_{k}$ is our new best estimation, and we can then feed it (along with $P_{k}$) back into another round of predict or update as many times as we like. The result using Kalman filter always outputs a smoothness trace and high precision at localization, since we adopt noise elimination and dynamic motion.

### 3.3. Localization Refinement with Dynamic Motion Model

To build a flexible motion model for indoor localization, we discretize motion data into step intervals instead of just recording at a particular frequency (e.g., every 0.5 s), so that the uncertainty elimination algorithm can achieve great performance. The accelerometer data are gathered and processed in real time. The implementation is intended to be device-agnostic, using the sampling rate of the sensor in the device. Ref. [52] gives a robustness motion model which can endure the unexpected orientation and step changes, but it is hard to implement in practice because of the high time complexity. In our model, we break the continuous motion into some common patterns. Formula (Equation 11) gives a brief description of the dynamic motion model.
(11)$$\left\{\begin{array}{c} X_{O}^{k+1}=X_{O}^{k}+\Delta X_{O}^{k}+\sigma_{w_{O}}^{k}\\ X_{P}^{k+1}=X_{P}^{k}+l_{step}\left[\begin{array}{c} cos(\Delta X_{O}^{k})\\ sin(\Delta X_{O}^{k}) \end{array}\right]+\sigma_{w_{p}}^{k} \end{array}\right.$$ where $l_{step}$ is the step length, $\Delta X_{O}^{k}$ is the user’s heading changes between *k* and k+1 step, and $\sigma_{w}^{k}$ is Gaussian noise. The step length $l_{step}$ is not a constant value in the motion model, and it is estimated dynamically during localization.

For motion discretization and steps separation, it is common to employ the zero-crossing algorithm [56] to divide the trajectory into *k* steps. A simple variety of the zero-crossing algorithm is implemented in the Android operating system to detect the number of steps. We know that the vertical acceleration signal crosses the zero line twice every step. In the experiment, we would decide whether it is precise enough to use in indoor localization.

Based on previous work [32], the following aspects are used to judge whether the user is in walking state:

(1) We only count the crests and troughs over the threshold that we set.

(2) There are specific maximum time intervals between two steps of the human body constrained by reflex nerve, so we filter the unrealistic narrow wave as noise.

(3) Precision also depends on device sensitivity. In real localization situations we have to make the frequency of the sensor uniform.

The orientation of the next step is determined by observation of the gyroscope indicator. After obtaining our estimation of current position, it is common to assume that we still have a lot of steps data which can help to calibration the localization result. We employed an enhanced pedestrian dead reckoning (PDR) to derive the complete state; refer to $X={\left[P,O\right]}^{T}$. Position *P* was already been calculated, so in the current step, the next orientation *D* should be then described with gyroscope data.

Similarly, the orientation would be modeled with uncertainty: $p\left(D_{k}\right)∼N\left(\Delta d,\sigma^{2}\right)$, where Δd can be derived from residual error of model estimation and observation, and the value of σ would be determined by model iteration, which will be discussed later. As we mentioned above, errors of gyroscope measurement also need to be included. For a specific estimate point in one step, all possible next positions with orientation form a fan-shaped area, whose center is the current position. The observation, however, has a higher possibility as the true orientation. The other orientations around it with a small-scope angle difference would have relatively lower possibilities.

The first state would affect the performance of GeoLoc as a whole, which means that initial parameters in *D*, such as angles and confidence level, can also be parameters which would be adjusted later.

Then, in the iteration phase, we should make sure that the angle of candidate orientations and radius are not increasing unconstrained. Thus, we need to adjust the noise and variance accompanied by walking. The formula below expresses this in detail.
(12)$$O_{0}=dO_{k}-\omega_{o_{k-1}}\times \Delta d<=O_{k+1}<=O_{k}+\omega_{o_{k-1}}\times \Delta d$$
where *O* is a vector of scalar radian value; for each item in *O*, it follows a different Gaussian distribution. It is easy to imagine that in one step the candidate orientations compose a fan-shaped area, with step length Δd as its radius and uncertainty $\omega_{o_{k-1}}$ as its angle.

In addition, this phase can automatic modify its uncertainty level. The idea of this method is to improve prediction performance by making full use of observation data in previous steps. Due to the convergence of this procedure, GeoLoc would keep iterating until a maximum or minimum threshold is reached [57]. When the uncertainty and also the candidates are too large, we just reject the amount over reasonable scope. Thresholds are hyperparameters so we can modify them in practice. The whole process is presented in Figure 3.

Note that $\omega_{o_{k-1}}$ is uncertainty following Gaussian distribution. We use the results obtained from the last two phases to adjust parameters in (Equation 13):(13)$$rectification_{k}=d\left(filter_{k},matching_{k}\right)$$
where $recification_{k}$ is the offset between observation and prediction in $K_{th}$ step, $filter_{k}$ is position after filter process, and $matching_{k}$ is the origin matching result position in the map. In this case, *d* denotes a function calculating Euler distance explained later. We apply this rectification to uncertain parameters σ in Gaussian distribution iteratively with Equations (Equation 14):(14)$$\left[\begin{array}{c} \sigma_{\omega_{p_{k}}}\\ \sigma_{\omega_{o_{k}}} \end{array}\right]\propto \frac{rectification_{k}}{rectification_{k-1}}\times \left[\begin{array}{c} \sigma_{\omega_{p_{k-1}}}\\ \sigma_{\omega_{o_{k-1}}} \end{array}\right]$$

Ideally, σ would descend progressively and converge to a constant, meaning that we generally believe our observation has some error with constant value, or even that no error occurs at all (Algorithm 1).
**Algorithm 1** Algorithm.1:**procedure**Building2:   parameters←initial value3:   map←geomagneticmap4:   stepData←trajectorydata5:**end procedure**6:**procedure**Matching7:   using *geomagnetic threshold*8:   **if** stepData[current].magnet **not in** map **then return** false9:   **end if**  **return**
$P_{1}\leftarrow$ possible position area10:**end procedure**11:**procedure**Filtering12:   Input *past trajectory*13:   Output *smooth trajectory, prediction*14:**end procedure**15:**procedure**Motion model16:   **for** step **in** stepData **do**17:     using *angle, speed in previous steps, add uncertainty*18:     **if** prediction positions by inference **not in** $P_{1}$
**then return** false19:     **end if**  **return** $P_{2}\leftarrow$
*intersection of prediction and $P_{1}$*20:   **end for**21:**end procedure**22:**procedure**Handling outlier positions and error23:   Input *each trajectories, estimation points*24:   Output *trajectory, point by correction*25:**end procedure**26:**procedure**Evaluation27:   Compute *euclidean metric between test path and train, using DTW to compare trajectory*28:   Compute *euclidean metric of final position*29:**end procedure**

### 3.4. Outlier Positions Detection and Error Handling

For one path, consider that the model’s estimation described above is out of expectation; some corrective mechanisms should exist, such as indicators, to judge whether the path is worth exploring or we should just abandon it. Error-prone problems are as follows:

1. In the prediction phase, results of orientation and position are out of the map boundary.

2. Distance from the current observation to the last one is much further than previous distances.

3. No point in the map can be matched.

4. Orientations have a totally different result area with matching results.

5. Misestimation of turning position. Thus, a series of points would be misjudged.

These items have high likelihood to occur, but assumption without them can guarantee a uncomplicated degree. While we leverage PDR and Kalman filter in this work, we considered manually adding some rules to overcome them. For question 1, we move the out-boundary point to the closest boundary point, then perform the next check and process.

Question 2 offers the choice to discard the point or the path, which means we use prediction only in this step, and see the result in the next step. A valid problem 2 is described below:(15)$${\hat{x}}_{k}-x_{k-1}\gg \frac{1}{k-1}\sum_{i=1}^{k-1}x_{k}-x_{k-1}$$

We change the threshold as a variable to solve problem 3, and if there is still no magnetic observation result, we just discard this candidate route. By employing the iterative uncertainty elimination algorithm, problem 4 nearly disappears; we can also add the uncertainty to obtain orientation. Problem 5 is hard to solve; we just include landmark as a primary plan. Some factors and problems are ignored here, but these items remain and should be quantized and applied as our future work.

When applied to smartphone tracking, motion estimation usually incurs much more noise than robots, such as step miscounting, step length estimation error, or change of heading offset (i.e., the difference between user heading and phone heading). These errors tend to lead to localization failure.

## 4. Experiment and Results

In this section, we illustrate the results of experiments on an implementation of the model, then analyze them. Misjudgments might come from every procedure, so experiments in this paper were designed to test them separately. After the whole procedure is finished, we use the sum of Euler distance in every step position from prediction to ground truth as error of the model. Our experiment results are considered under other result evaluation methods, which are mentioned in the evaluation part, which may have a different performance, so we compare these indicators as well.

For our experiments design, we propose the following issues that experiments should focus on: (1) To what degree does the different user trajectory influence GeoLoc performance? (2) How many motion estimation errors can GeoLoc tolerate? (3) How well do the landmark and different magnetic fingerprint maps perform? (4) How well does the iteration algorithm perform and monitor converge status? (5) How well does the algorithm perform under different parameter situations? (6) Performance comparison of GeoLoc with different types of localization models.

### 4.1. Experiments Preparation

In the evaluation phase, we applied the Euclidean metric in GeoLoc. Notice that GeoLoc returns a whole trajectory, so error of the whole trace can be calculated as an important indicator. This means that we consider all points we estimated, using mean value of error of them, shown in (Equation 16). Another indicator would simply be the distance between ground truth and estimation value of the last position. We employ the two in the following experimental part.
(16)$$d={∥\sum_{k=randomChoice}^{candidateSet}GroundTurth_{k}-Estimate_{k}∥}_{2}$$

We chose several numbers of trajectories from the candidate set randomly as our final estimation. The mean error of these trajectories would be our evaluation result. In addition, we sum up every position’s distance between real trace and estimation, which are also obvious indicators.

We also employ the dynamic time warping (DTW) method as an option evaluation method. DTW is a well-known technique for aligning two time series sequences with similar patterns but with amplitude and time differences. It has been widely applied in speech processing, sensor data classification, and data mining [51]. The magnetic signatures collected by steps can be classified as time series data, because they were collected at certain time intervals. When we walk along the similar trajectory, we would observe that the signatures follow a similar pattern. We can use that pattern to compare the test traces, so a huge difference between two signatures means a huge error in this localization process. The ground truth pattern can be generated from the magnetic map we built, then our results—candidate trajectories—are compared with it. Figure 4 shows the similarity of five different paths of a same corridor. We can see that if the corridor is divided into five paths, we can clearly see that the closer the path is, the closer the trend of data is. In the same way, even if the path distance is far away, the similarity of the data is considered to be different.

### 4.2. Performance of Building Fingerprint Magnet Maps

The feature of this article is to realize the function of accurate indoor positioning without relying on the basic environment, so we have to use the technique of location fingerprinting, and the invariance of the data is the most important for fingerprint position. Geomagnetic data are very suitable for this application scene, and can be used as fingerprint data. Because, in the indoor environment, buildings and some internal facilities are relatively fixed, and the effect on the Earth’s magnetic field is also relatively fixed, the geomagnetic data are very stable, and will not change with time, as shown in Figure 5.

We collected magnetic data in Northeastern University’s Information Building on the fifth and sixth floors, and examples are shown in Figure 6. The building structure is mainly reinforced concrete, which offers a stable and characteristic magnet distribution. Inside the building, there are small floorsand adjacent corridors in these floors, as classic geographical features. We segmented the test ground into several square grids, to generate a detail-oriented magnetic map. Figure 6 was collected in an indoor environment with an area of about 6.4 m × 6.4 m in the location of the information building, and the data are relatively sparse. After using the interpolation method, the density of the data is improved, which greatly reduces the pressure of data acquisition.

It is generally accepted that, although stable enough, magnetic values still have inconstancy to some extent. One reason is that the magnetic field would change over time. Thus, we first collected data 10 times during 1 month, taking the average of magnetic values in each position. Afterwards, we obtained several versions of the magnetic map. The physical relative location was also recorded to tag the sensor readings with their corresponding location, under a coordinate system we set. These maps have the same tendency in general, and each data point fluctuates around the mean value.We used sample mean as real magnet value in corresponding positions, under the assumption that the confidence interval of the mean value can guarantee a relatively precise magnetic value.The reliability of the magnetic map was also attested.

Another variation of the magnetic map is using a transformed *Z*-axis of magnetic value [55]; since the transformation readings along the *Z*-axis are independent of the phone or the user’s orientation and gesture, we can apply them as magnetic information for each location. The transformation mainly uses the concept of Euler angle, by projecting the 3D magnetic readings onto the world’s *Z*-axis provided by the gravity vector (three-axis accelerometer) obtained previously.

There are well-known methods for measuring the direction of gravity based on accelerometer and gyroscope sensors, such as combining the information from gyroscope and accelerometer (usually using a Kalman filter) to obtain more accuracy and less delay. We used the APIs already implemented on Android phones [58].

Correspondingly, magnetic readings in the step trajectory also need to be transformed into *Z*-axis format. We used the comparison test to verify the reliability of this approach. We tested the validity of *Z*-axis transformation by comparing it with quadratic mean of magnetic value, calculating DTW distance as a benchmark. Magnetic track data using data from all axes are shown in Figure 7. It can be observed that only an offset would affect the projection on the *Z*-axis, while the overall magnetic values are not constant. The calibration effects are shown in Figure 8. Using DTW to measure the difference between the two sequences, we can conclude that the data after projection to *Z*-axis would achieve a higher precision, because test cases match the pattern of the template.

Both the two methods of magnetic construction, sum of square and transformed *Z*-axis value, were implemented, and we used algorithms comparison to evaluate these methods.

Due to the different mobile phones having some inherent differences, such as the frequency of adoption rate, the algorithm must have the ability to adapt to a variety of models of mobile phone. We used different smartphones at the same location to measure the results. The results are shown in Figure 9. Although the results of different mobile phone measurements are different in value, the data difference is stable in general. Based on the above findings, we believe that a certain threshold can be used to adapt to different mobile phones when the geomagnetic data are matched.

Because of the different users’ height, users may also place their phones in any pocket. In other words, the height of users holding mobile phones is different. Therefore, it is necessary to explore the influence of the height of the mobile phone relative to the ground on the geomagnetic value. We tested the change of geomagnetic value at different heights at the same location, and the test results are shown in Figure 10.

From Figure 10, we can see that when the smartphone is at different heights from the ground, the geomagnetic value is changed, and the geomagnetic data from 1 m, 0.5 m, 0.3 m, and 0 m are different. Compared with the 0.3 m and 0 m location data, the data from the 0.5 m location data and the 1 m location data have higher similarity, so we believe that the lower the height difference at one location, the higher the similarity of the geomagnetic data.

### 4.3. Evaluation of Step Trajectory Pattern

Then, step data are collected during user walking. In the step dataset, errors would also be accumulated via step counter process and calculating process. This means that the trajectory data should be preprocessed by step before being applied.

To build the step trajectory dataset, we walked many paths in the whole area, randomly picking initial positions of the building with a smartphone in hand. The observation trajectory information can be calculated via different methods, such as several checkpoints given by step counter along the paths, while assuming constant walking speed between each two points. The target of this phase was to obtain clear data of every step, including steering angle.

In [59], a turning detection model was employed to make localization system independent ofthe gesture. To test the effectiveness of GeoLoc in complex motions, we planned some daily motion paths as our test cases. Supposing the path in real localization problems would be regular, we used straight segments with no, one, and two turnings, representing motion patterns regarding walking straight and turning around. The robustness of GeoLoc in turning cases is shown with CDF in Figure 11.

Evaluation methods using all points in the path show that the more turns the user takes, the more errors occur in all algorithms. GeoLoc has the advantage that the estimation of the final position is relatively better than the normal Kalman filter.

### 4.4. Effect of Errors Using Motion Model

We need to process step data before we use them. When walking along a hallway, the data we collected are continuous. We have several methods to discretize our data, to make them suitable for different algorithms. A common method is dividing them by a specific time slot, and dividing by step gives us convenience to use PDR, because PDR is a discretized algorithm using relative position. Therefore, we focus on step counting methods to discrete step data.

Step counting is the foundation of the motion model. If there is a large cumulative error in step counting, the effect on positioning process in later stages can be disastrous. The above method also shows a better step precision, as shown in Figure 12. The transverse axis in the diagram represents the true steps of the walk, and the longitudinal axis represents the step precision. It can be seen from the diagram that the algorithm can reach a precision of more than 90%, and it is sufficient to establish the PDR model in the room. The Meizu mobile phone, which has a higher frequency of data, will show a better effect than the Samsung mobile phone.

By comparing the performance before and after the use of the step counter, it is obvious that our step counter model has enough precision to be used in indoor localization. The step counter also reduces the misjudgment of PDR in the next phase.

The estimation of the step length has always been a very difficult subject. As the step is constantly changing in the course of walking, this article will constantly modify the step length in the process of positioning. The step length is related to the speed and frequency of human walking, so it is also associated with the accelerometer. The specific length of each step can be estimated from the readings of the accelerometer. The above experimental data were mainly collected by the same person, so acceleration variation at different step sizes are valid for step length estimation.

Figure 13 shows the change of the acceleration after the user walks in different steps. The figure with an asterisk identifies the change step of 60 cm acceleration, and the bar logo showsthe change in acceleration to 80 cm step. It is obvious from the diagram that the larger the step size is, the greater the amplitude of the accelerometer is.

Some localization systems [59] and pedometers build a complicated relationship between a motion model and step length. Therefore, in GeoLoc, we also suppose that different users have different walking styles. Formula (Equation 17) directly gives the rules found in the paper [51] according to many experiments. N represents the number of samples collected in a walking cycle, while A represents the accelerate value at sample k. To use the formula N, weneed calibration first. This formula establishes the relationship between the accelerometer and the step size, and the next work is based on this conclusion.
(17)$$Stride\left(m\right)=0.98*\sqrt[{3}]{\frac{\sum_{k=1}^{N}\left|A_{k}\right|}{N}}$$

The specific error results after calibration are shown in Figure 14. The *X*-axis in the graph is represented in six different scenarios. Y represents the difference between the estimated step length and the actual step length, and the unit is represented by M. The upper and lower two edges of each box in the graph represent the maximum and minimum of the error. The upper and lower edges of the rectangle in the box represent 75 and 25 sub-positions, respectively, and the middle line in the rectangle represents the middle error. In addition, there are several plus signs above each box to indicate the exception value in the estimation process. It can be seen from the diagram that the difference between the estimated step length and the actual step length is very small. Even if some exceptions are taken into account, the error of the step length is negligible.

Comparison tests of robustness among different models are illustrated with CDF in Figure 15. The figure shows the cumulative distribution 16 and orientation 5. Exemplar tracks were obtained by the same person from the same walking path. We can see that the localization accuracy fluctuates with the initial steps, in Figure 16, then the similar pattern is shown. If the geomagnetic threshold is too small, it is hard for the pure PDF method to obtain a high localization accuracy, while GeoLoc can achieve high precision rapidly. The average localization error can decrease to 2 m in 80% of situations when the step modelis introduced.

### 4.5. Performance under Different Parameters and Landmark

Indoor localization models require a rapid precise positioning result after necessary short movement. Filtering and matching of the algorithm’s complexity also restrict the longest path we walked.Alongside the adjusting of other parameters in our algorithm, we tested the system’s performance under different lengths we walked. For an intuitive description of these tests, floor plan and trajectory localization are listed in Figure 17 and Figure 18. It shows that converging in a corridor is faster than the turning paths, and the more turning there was, the more error that occurred, since turning action would temporally decrease localization accuracy. However, if the turning was obtained correctly, the localization process would speed up and increase the precision.

Then, we can choose a proper value as threshold of length under these parameters’ combination, according to Figure 16. It comes down to relationship between errors and length of walking. Total lengths less than 5 m have a huge change of accuracy because landmark in this case has the most important weight above all factors, and it has a limited length because of the physical limits of the map. For total step length above 5 m, the decreased tendency is obvious. The minimum number is around 5.5 m, which means that the balance of prior knowledge and predict information have been achieved.

To analyze the mechanism, here we consider amount of error reducing during certain steps as converging speed. Matching phase is performed as checkpoints along the path, to correct the positioning result. Due to the amount of information used, the precision of the two are different. After the data collection phase is finished, all information could be used; in addition to more accurate results, the two problems described above can be solved to some extent.

For the iteration phase of the algorithm, there are circumstances that no results or no reasonable results produced. This is due to the possibility of not converging at all, brought by the algorithm’s uncertainty. Even in most cases, the final estimation path converged to the correct path. That means that GeoLoc does not guarantee certain localization accuracy in rare cases. A common case leading to early convergence is that matching results provide regular and useful prior information, while convergence fail chance can be significantly increased if using matching disturbance prior information.

With regard to magnetic matching threshold and other parameters in different places, we tested GeoLoc in various indoor environments. In GeoLoc, the matching phase was most affected because of the widely distributed magnet value of different physical characteristics. In other words, we can obtain more accurate matching results rapidly by only setting a relatively small threshold of matching scope. For example, we built magnet fingerprint maps for pillars, vendor machine, and corridor with pillars, as shown in Figure 19. Significant distinctness appeared among these magnetic maps; we can see that there are many points with the same geomagnetic field strength in normal ground and distinctive magnetic field of these physical features as landmarks [9]. Note that in the map, the valley around x=0 represents the building wall with elevator. In the *X*-axis section, the shape is obviously a plateau, which means that it can be identified out of other areas. Previous work [9] used a landmark to make the matching process more expeditious and accurate. These building features can generate a discrimination map for localization. Therefore, in the *Y*-axis section, a common pattern of magnet value corresponding to building features appeared. Specifically, there are two elevator doors in position at about y=15 and y=9, a pillar in y=2, and they have ridges in the map separately. The vendor machine represents a valley in the magnetic map. x=0 has the most difference because it is the nearest. In y=8, there are steel doors, so a peak appears.

Figure 20 shows the distribution of average localization errors of our model under different landmarks, with the same parameters, such as geomagnetic threshold and walking pattern. We can see that the 80th percentile error of landmarks was within 1.2 m, and the value of normal place was within 1.6 m. From the figure, we can conclude that error around landmarks is nearly equal to normal ground. We can conclude that the test path along these landmarks tends to give a more reasonable and accurate result.

### 4.6. Overall Performance Evaluation

We then evaluate the overall performance of GeoLoc. The problems remaining are mainly cold start and error accumulation problems, which commonly occur in localization models. Here, we use the evaluation method with final point measurement, called point distance error. In the experiment shown in Figure 18, we apply the measurement method to every point.

Cold start problems concern the issue that the model cannot draw any estimations for users or items about which it has not yet gathered sufficient information [60]. Results shows that due to the help of matching and landmark, 72% of misjudgment in the first five steps are under average distance error.Error accumulation occurs in the later stage of trajectory; for data sources, errors in the gyroscope sensor will result in large accumulated errors in estimation; for models, values of threshold and parameters lead to underfitting and overfitting, which is another main reason for accumulated error. Figure 21 shows the precision using the GeoLoc algorithm to locate in different paths. It can be seen from the graph that the location accuracy of this algorithm is relatively low at the beginning because of the cold start problem, and the error is about 5 m, which means that at the initial time, the accuracy of acquiring initial position by geomagnetism is not high, and the algorithm hascold start. With the continuous movement of users, the accuracy of positioning is increasing. When the user moves around 20 m, the positioning error falls to about 1 m. The algorithm in this paper has similar trends in the location accuracy under multiple paths, and most of the errors are within 2–4 m, so we can say that the algorithm is relatively stable. Therefore, this article is effective for the improvement of the PDR model.

In this paper, the particle filter algorithm is also included in the comparison of the algorithm experiment. In the experiment, we compared the PDR algorithm, the GeoLoc algorithm, and the particle filter algorithm proposed in this paper, in which the particle filter was implemented using 1000 particles and 5000 particles, respectively [61].

The results were obtained in a finite number of steps (10 steps) tested along the corridor, and the path was linear. Experimental results are shown in Figure 22. It can be seen that GeoLoc has the smallest positioning error and faster convergence than other methods.

The contrasting results are shown in Figure 23. The *X*-axis of the graph shows the distance of the location, and the *Y*-axis indicates the error of the positioning. From Figure 23, we can see that the positioning error of the PDR model increases with the increase of distance, and the location error of the algorithm in this paper is decreasing continuously. It can be seen from the diagram that the error of using 5000 particles is less than the use of 1000 particles. In comparison, the accuracy of this algorithm is higher than that of the particle filter in most cases. That means that the more complex the model is, the better the result is. Only a suitable model can obtain better results.

In terms of location timeliness, a particle filter is originally used in Monte Carlo. If the number of particles is too high, it will greatly affect the efficiency of the algorithm. Figure 24 shows a box map of the time complexity of the location. The *X*-axis represents the selected algorithm, and the *Y*-axis indicates the time of the algorithm. From Figure 24, we can see clearly that the time spent in the PDR algorithm is determined because of the calculation process, and the time complexity is also increasing with the increase of particle number. The time complexity performance of the algorithm presented in this paper is far more than the particle filter algorithm.

## 5. Conclusions

In this paper, we present an indoor localization model using geomagnetic data on a smartphone platform. The indoor environment is complex and turbulent, and inertial measurement unit (IMU)-based systems require a simple time complexity, so our model is capable of distinguishing user trajectory with a high accuracy. This is owing to the fact that our algorithm is robust with respect to a deviation in step size or orientation estimation. For the error caused by human steps, we employed an augmented pedestrian dead reckoning (PDR) motion and step model, reducing median error by about 0.8 m. Compared with the method only employing PDR or filter, our model solved the cold start and accumulated error problem to some degree. The experimental results also show that using only *Z*-axis data would give us a better precision. Our model aimed to perform in a user-friendly localizationwhere the user would be required to walk only a short distance, and our test results returned a satisfactory precision of around 5.5 m walking length, with precision around 2 m achieved.

This work only uses part of the sensors in smartphones, and our system requires prepared fingerprint data, so it is very time-consuming in the real world. In the future, we will consider integrating more built-in sensors of smartphones to improve accuracy, and designing new fusion algorithms to improve the accuracy of indoor positioning. In addition, simultaneous localization and mapping (SLAM) framework or crowdsourcing methods are used to construct the geomagnetic fingerprint map. Future work will also involve evaluating our model in a larger number of buildings to find potential parametric patterns, such as thresholds for matching phases.

## Figures and Tables

**Figure 1 sensors-22-09032-f001:**
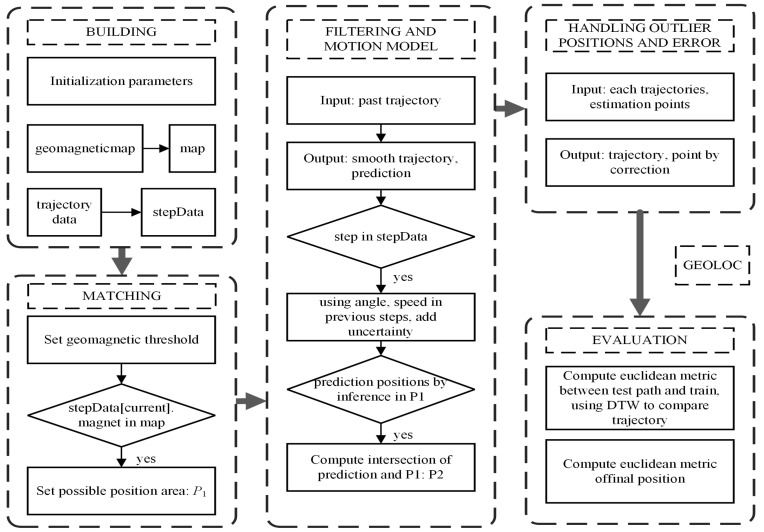
The model structure of GeoLoc.

**Figure 2 sensors-22-09032-f002:**
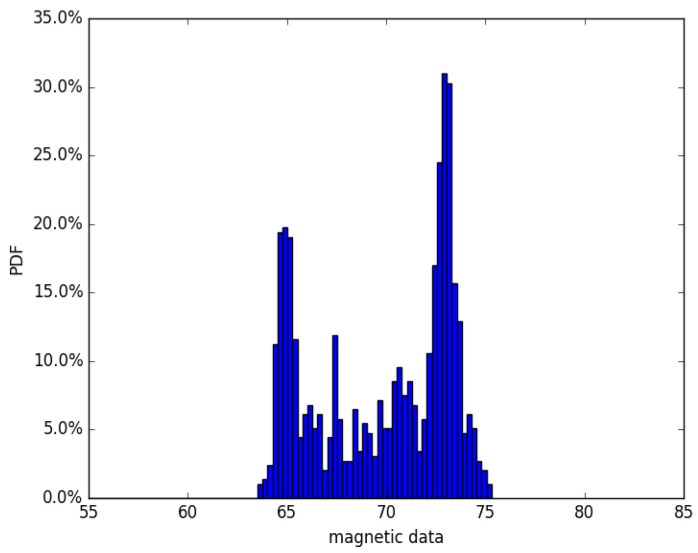
Walking recorded in a straight line with corresponding magnet value. It can be seen that magnetic values around 65 μT and 73 μT have more possibility to be observed. Observation near these values would introduce errors to estimation.

**Figure 3 sensors-22-09032-f003:**
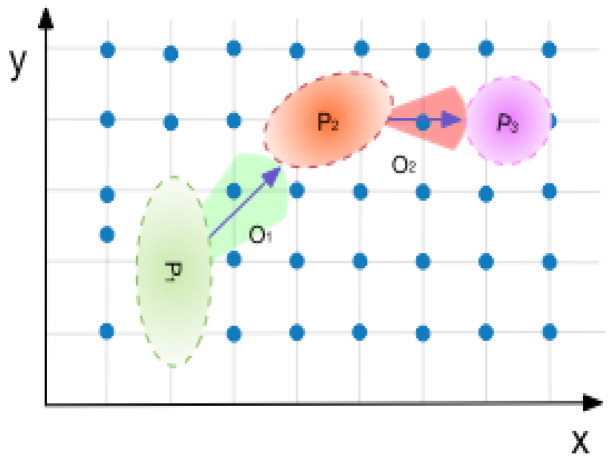
Uncertainty eliminates algorithm diagram explanation. From left to right is the user’s motion estimation. The colors of the fan-shaped area and circle area denote the uncertainties. In each step, the searching space, acreage in the figure, is diminished.

**Figure 4 sensors-22-09032-f004:**
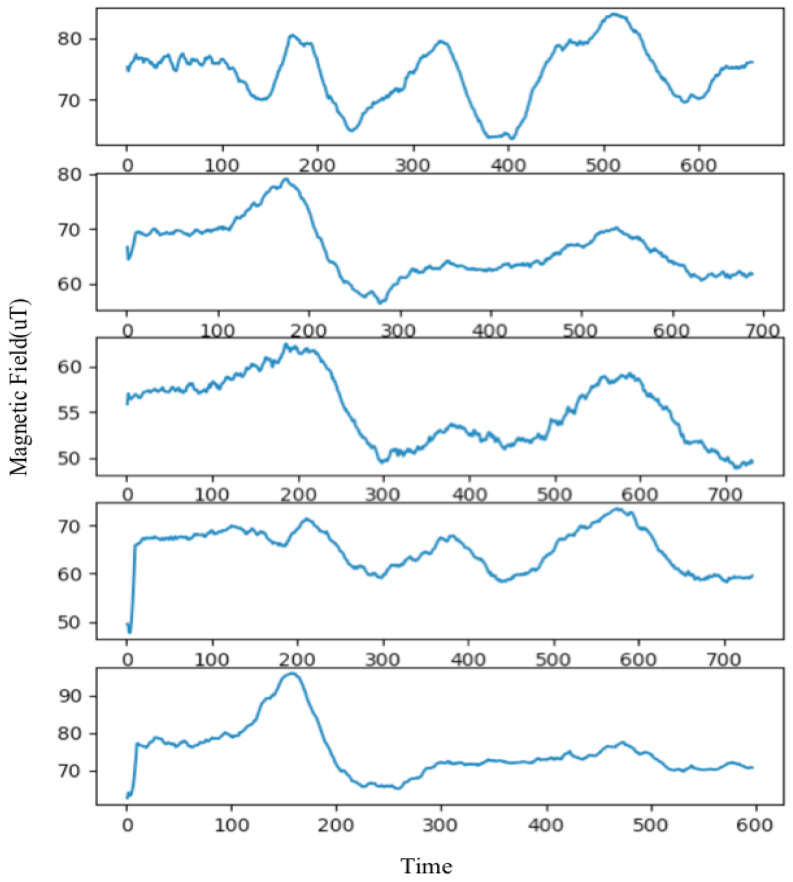
Five different paths of the same corridor.

**Figure 5 sensors-22-09032-f005:**
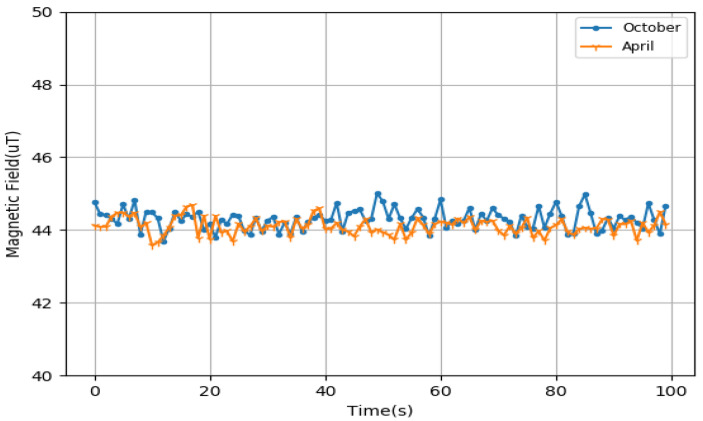
Magnetic field data at different times.

**Figure 6 sensors-22-09032-f006:**
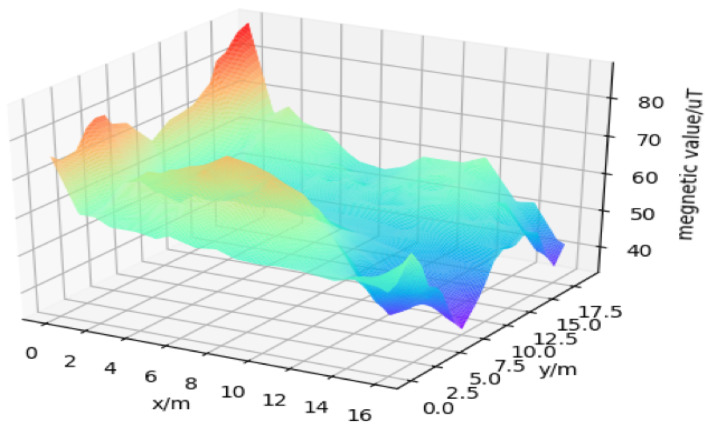
Example magnetic fingerprint map of the 6th floor.

**Figure 7 sensors-22-09032-f007:**
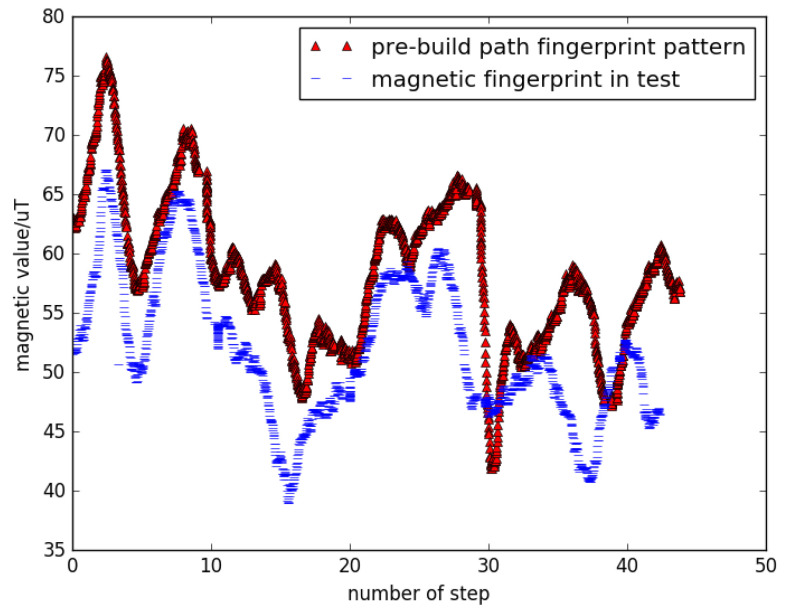
Magnetic trajectory data using all axes’ data.

**Figure 8 sensors-22-09032-f008:**
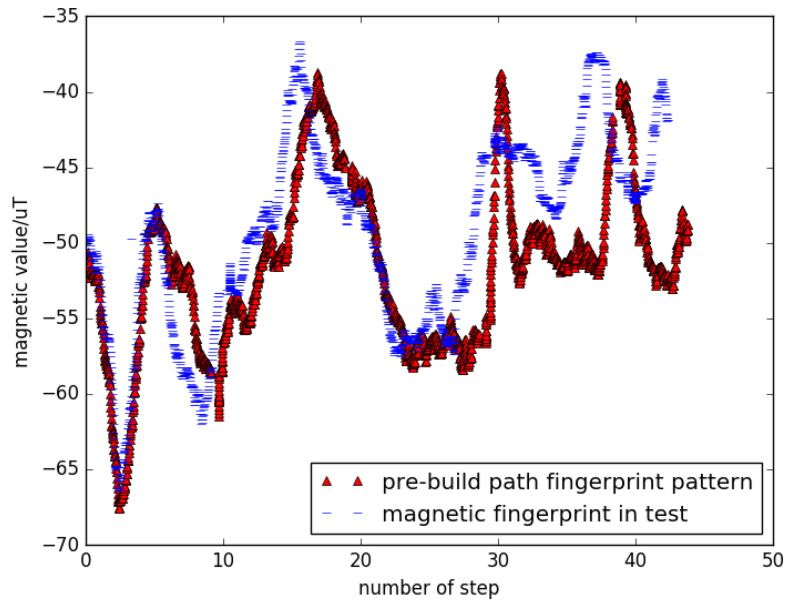
Magnetic trajectory data only using transformed *Z*-axis value.

**Figure 9 sensors-22-09032-f009:**
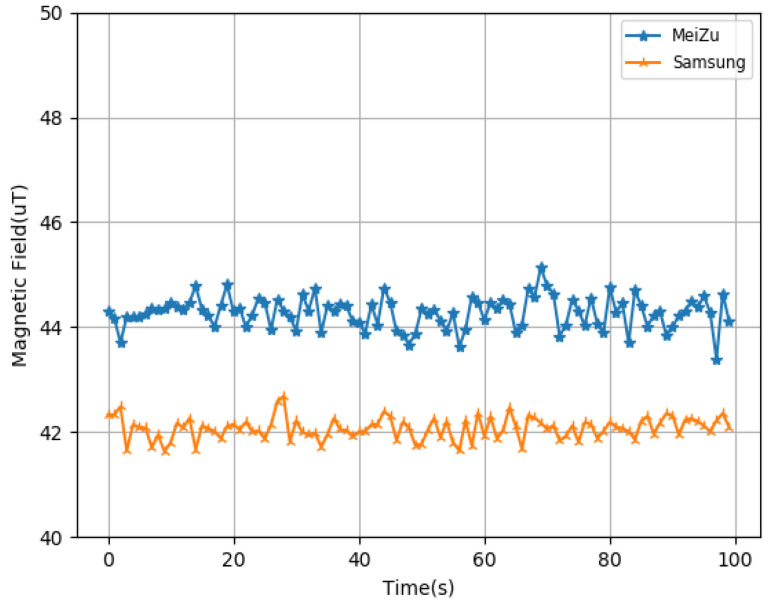
Geomagnetic data of different mobile phones in the same position.

**Figure 10 sensors-22-09032-f010:**
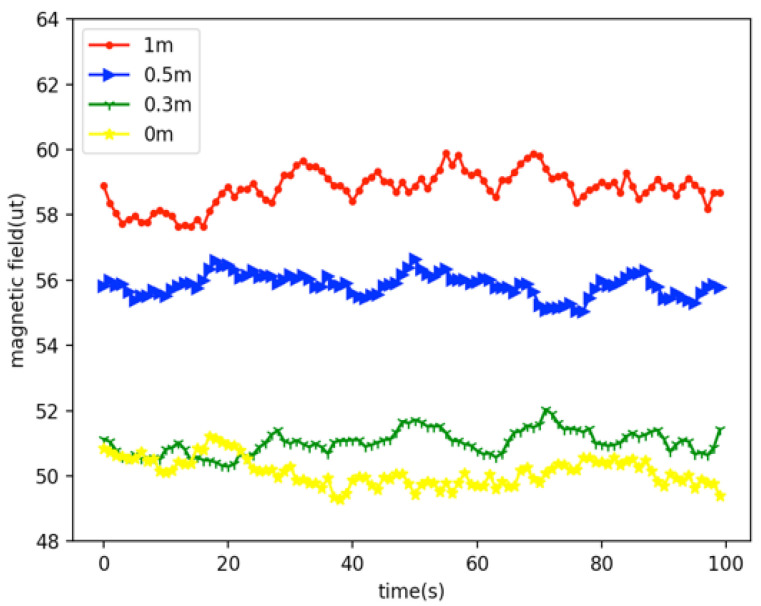
Geomagnetic data at different heights from the ground at the same location.

**Figure 11 sensors-22-09032-f011:**
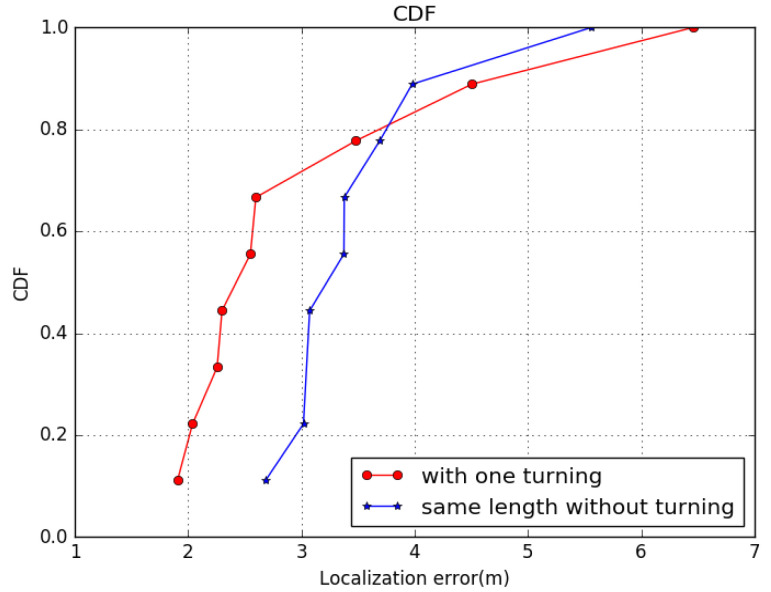
Figure shows CDF with localization error. The left curves represent steps with turns, the right denote the track at the same length without any turn. We can see that 80% of the errors occurred in less than 2.5 m.

**Figure 12 sensors-22-09032-f012:**
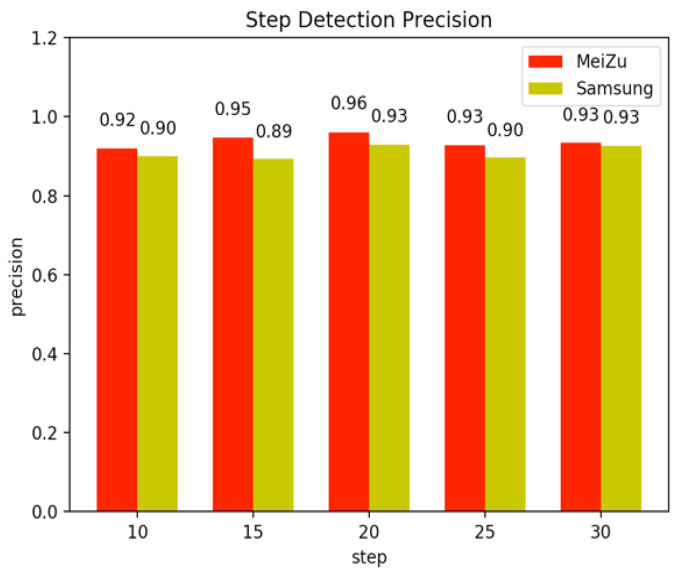
Step detection precision histograms under five repetitions.

**Figure 13 sensors-22-09032-f013:**
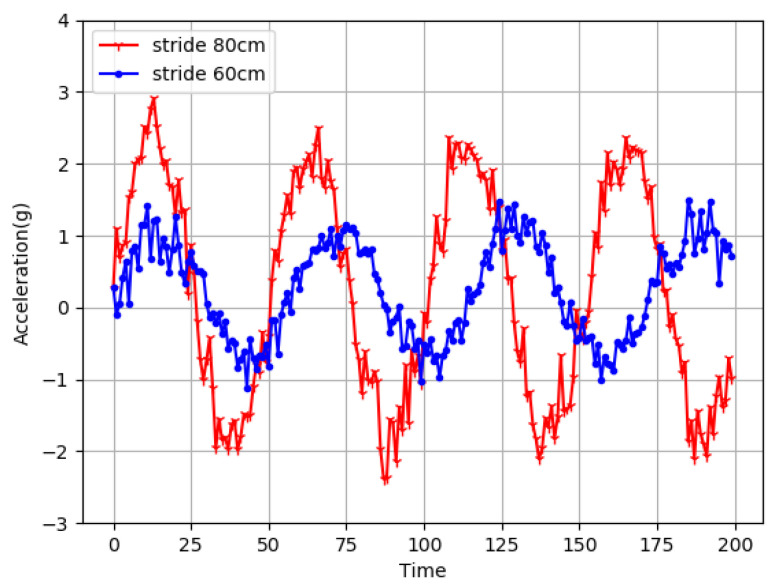
Acceleration variation at different step sizes.

**Figure 14 sensors-22-09032-f014:**
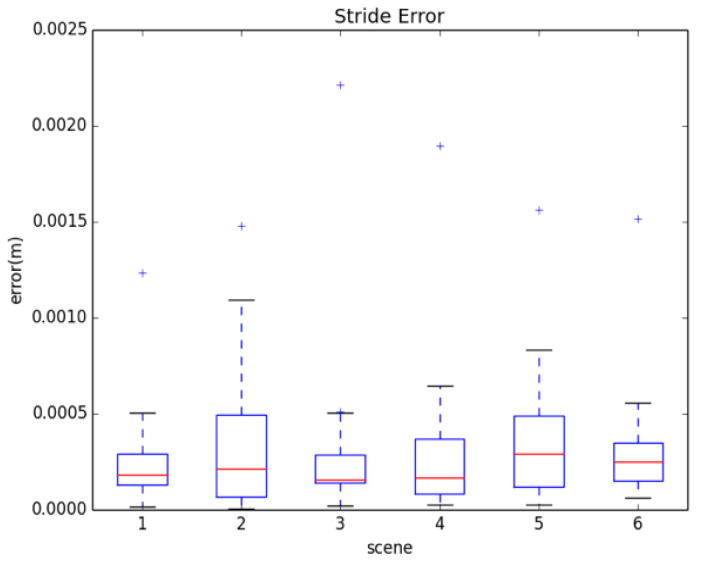
Box plot of stride estimation error.

**Figure 15 sensors-22-09032-f015:**
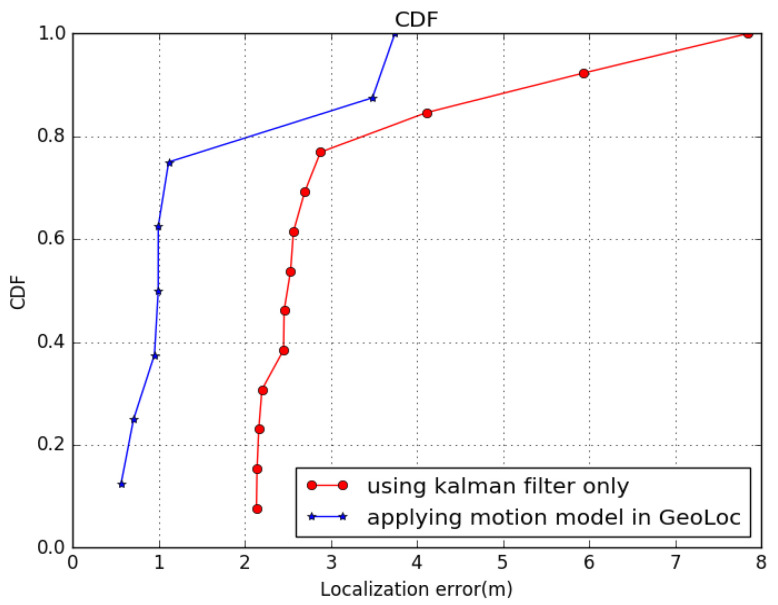
CDF among different models, plain Kalman filter, and GeoLoc.

**Figure 16 sensors-22-09032-f016:**
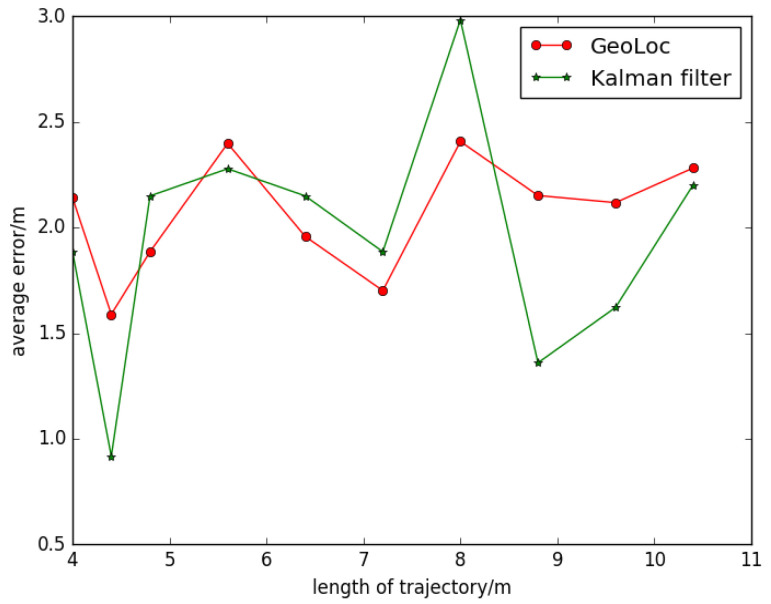
Average error varies with the length of trajectory change.

**Figure 17 sensors-22-09032-f017:**
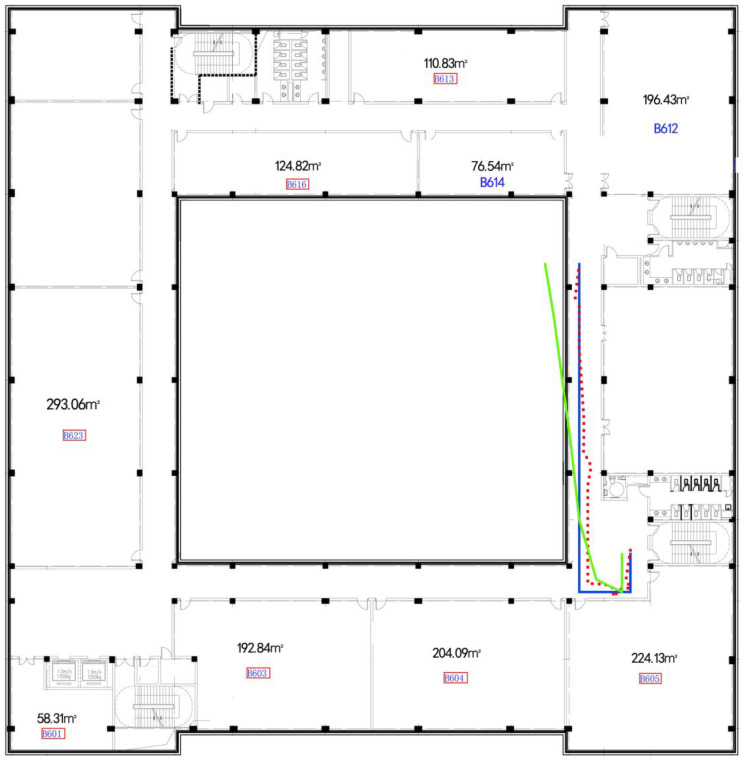
Floor plan of 5th floor.

**Figure 18 sensors-22-09032-f018:**
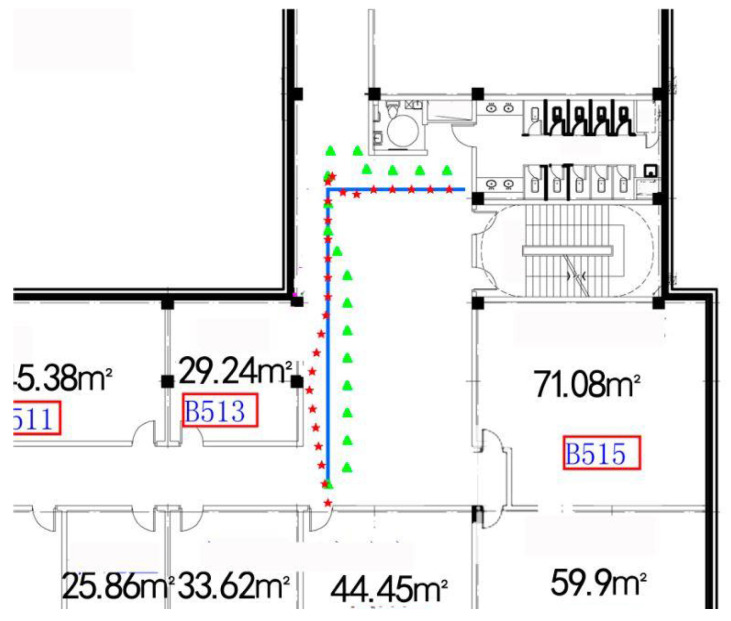
Floor plan of 6th floor, with estimation and true trajectories. The triangle line, or the polygonal line, represents normal Kalman-filter-based localization; the straight line with regular turning represents the test walking path, and the line with the red five-pointed star shows the results of GeoLoc.

**Figure 19 sensors-22-09032-f019:**
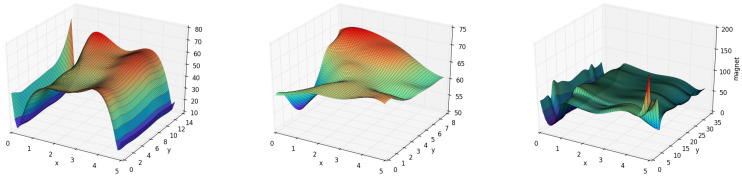
Magnetic map near landmarks: lift, vendor machine, and corridor, separately.

**Figure 20 sensors-22-09032-f020:**
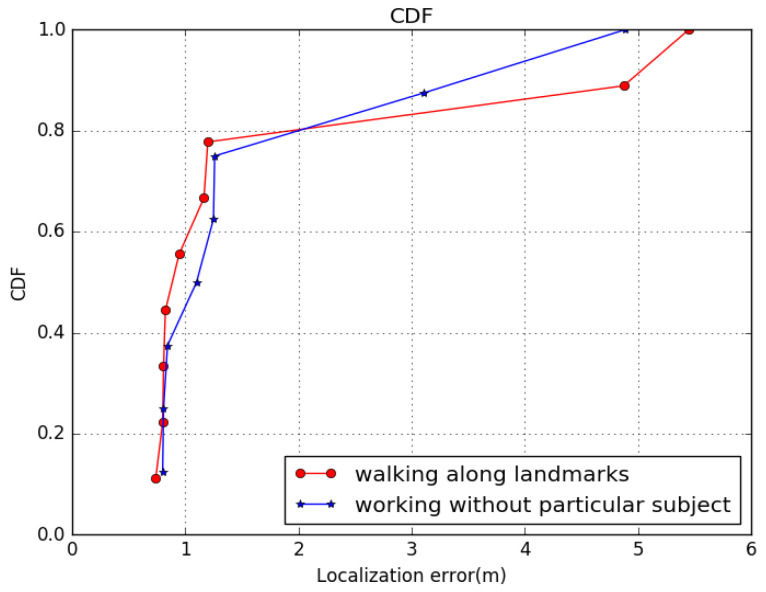
Comparison CDF of localization results around landmark. GeoLoc still has a slight advantage.

**Figure 21 sensors-22-09032-f021:**
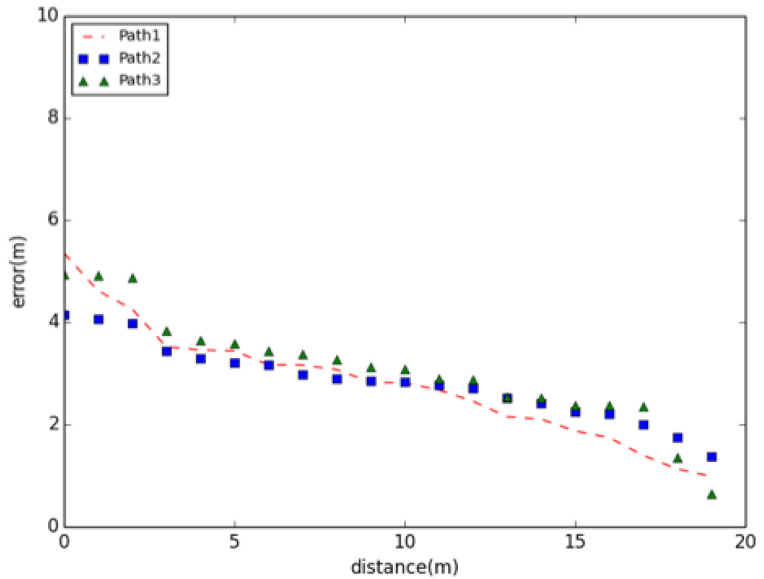
Precision of GeoLoc on different paths.

**Figure 22 sensors-22-09032-f022:**
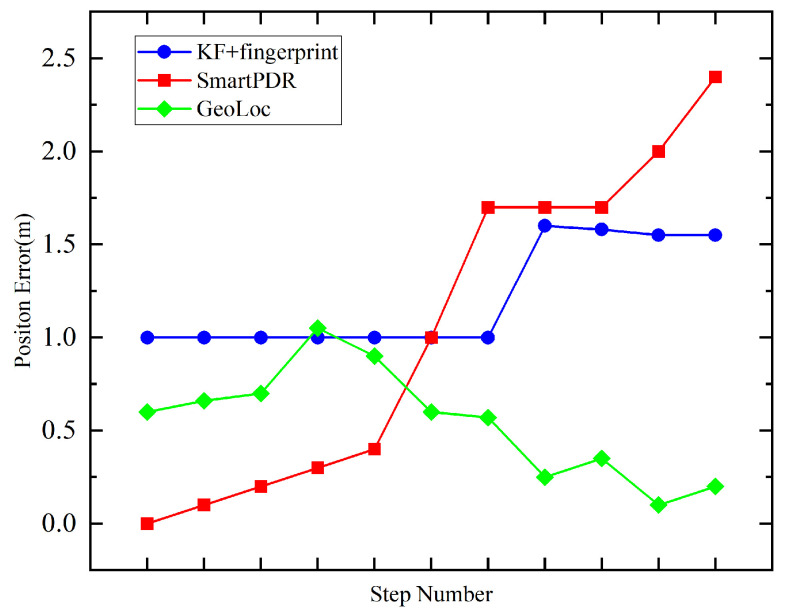
Relationships between step number and positioning error using different positioning methods.

**Figure 23 sensors-22-09032-f023:**
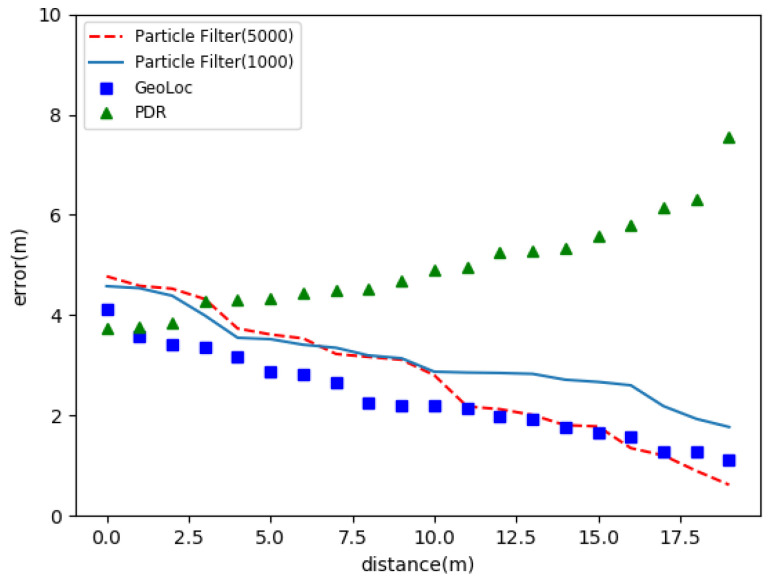
Comparison experiment of positioning accuracy.

**Figure 24 sensors-22-09032-f024:**
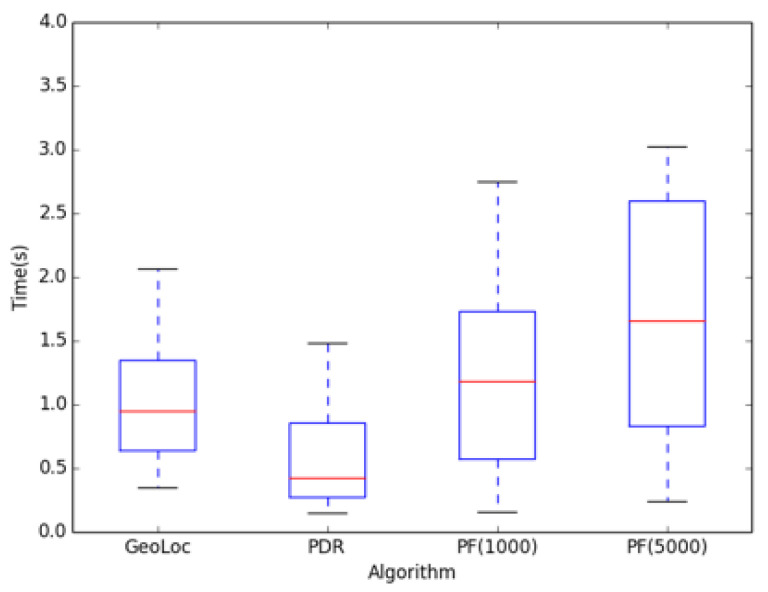
Comparison experiment of localization time.

## Data Availability

Not applicable.

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
