# Peer review of "An Algorithm with Iteration Uncertainty Eliminate Based on Geomagnetic Fingerprint under Mobile Edge Computing for Indoor Localization"

_sensors, 2022, doi:10.3390/s22239032_

Round 1

Reviewer 1 Report

I have attached my comments.

Reviewer 2 Report

This paper is generally well-written. I have the following major comments:

1) Please summarize the novelty and contributions point-by-point in the introduction.

2) Please point out future work in the conclusion

3) Please cite the following important paper:

An Indoor Environment Sensing and Localization System via mmWave Phased Array

Reviewer 3 Report

 The manuscript described an indoor localization system named GeoLoc which uses Kalman filter to fuse the data of the PDR and magnetic fingerprint.  Some comments on this manuscript are as follows:

1.In section 3 "Algorithm Design", the method of fingerprint matching and the creation of fingerprint database should be more detailed.

2.In line 573, what is it mean “step trajectory also needs to transform into Z axis format”?

3.In line 593, “at the same height” should be at different heights.

4.In section 4, these experiments seem to be completed in very simple scenes, which lack statistical significance. It is suggested to do the experiments in complex scenes, and use sufficiently long experimental paths and duration. Moreover the experimental data should be described in more detail, such as:

(1) the length, the duration of time, the steps and the path of the trajectories, the magnetic fingerprint database information in Figure 10, 14, 19 and 22.

(2)the number of repeated experiments in Figure 11.

(3)the speed of walking in Figure 12.

(4)the information of scenarios in Figure 13.

Round 2

Reviewer 1 Report

The authors answered all comments and modified the manuscript.

Reviewer 2 Report

No comments